# Minimally Traumatic Cochlear Implant Surgery: Expert Opinion in 2010 and 2020

**DOI:** 10.3390/jpm12101551

**Published:** 2022-09-21

**Authors:** Vedat Topsakal, Sumit Agrawal, Marcus Atlas, Wolf-Dieter Baumgartner, Kevin Brown, Iain A. Bruce, Stefan Dazert, Rudolf Hagen, Luis Lassaletta, Robert Mlynski, Christopher H. Raine, Gunesh P. Rajan, Joachim Schmutzhard, Georg Mathias Sprinzl, Hinrich Staecker, Shin-ichi Usami, Vincent Van Rompaey, Mario Zernotti, Paul van de Heyning

**Affiliations:** 1Department of Otorhinolaryngology, Head and Neck Surgery, Brussels Health Campus, University Hospital Brussels (UZ Brussel), Vrije Universiteit Brussel (VUB), 1090 Jette, Belgium; 2Department of Otorhinolaryngology, Head and Neck Surgery, Antwerp University Hospital (UZA), University of Antwerp, 2610 Edegem, Belgium; 3London Canada Health Sciences Centre, Department of Otolaryngology–Head & Neck Surgery, Western University, London, ON N6G 2M3, Canada; 4Ear Science Institute Australia, Subiaco, WA 6008, Australia; 5Vienna ENT University Hospital, 1090 Vienna, Austria; 6UNC Ear & Hearing Center at Chapel Hill School of Medicine, Chapel Hill, NC 27599, USA; 7Royal Manchester Children’s Hospital, Manchester University Hospitals NHS Foundation Trust, Manchester Academic Health Science Centre, Manchester M13 9WL, UK; 8Division of Infection, Immunity and Respiratory Medicine, Faculty of Biology, Medicine and Health, University of Manchester, Manchester M15 6JA, UK; 9St. Elisabeth Hospital, Ruhr University Bochum, 44787 Bochum, Germany; 10Würzburg ENT University Hospital, 97080 Würzburg, Germany; 11Madrid Hospital La Paz, 28046 Madrid, Spain; 12Department of Otorhinolaryngology, Head and Neck Surgery “Otto Körner”, Rostock University Medical Center, 18057 Rostock, Germany; 13Bradford Royal Infirmary Yorkshire Auditory Implant Service, Bradford BD9 6RJ, UK; 14Luzern HNO-Klinik Kantonspital, 6000 Luzern, Switzerland; 15Department of Otorhinolaryngology-Head and Neck Surgery, Medical University of Innsbruck, 6020 Innsbruck, Austria; 16St. Pölten University Hospital, 3100 St. Pölten, Austria; 17Department of Otolaryngology-Head & Neck Surgery, University of Kansas Medical Center, Kansas City, KS 66160, USA; 18Department of Hearing Implant Sciences, Shinshu University School of Medicine, Matsumoto 390-8621, Japan; 19Córdoba Sanatorium Allende, Córdoba 5000, Argentina

**Keywords:** electric acoustic stimulation (EAS), cochlear implants, atraumatic surgery, hearing preservation, partial deafness treatment

## Abstract

This study aimed to discover expert opinion on the surgical techniques and materials most likely to achieve maximum postoperative residual hearing preservation in cochlear implant (CI) surgery and to determine how these opinions have changed since 2010. A previously published questionnaire used in a study published in 2010 was adapted and expanded. The questionnaire was distributed to an international group of experienced CI surgeons. Present results were compared, via descriptive statistics, to those from the 2010 survey. Eighteen surgeons completed the questionnaire. Respondents clearly favored the following: round window insertion, slow array insertion, and the peri- and postoperative use of systematic antibiotics. Insertion depth was regarded as important, and electrode arrays less likely to induce trauma were preferred. The usefulness of dedicated soft-surgery training was also recognized. A lack of agreement was found on whether the middle ear cavity should be flushed with a non-aminoglycoside antibiotic solution or whether a sheath or insertion tube should be used to avoid contaminating the array with blood or bone dust. In conclusion, this paper demonstrates how beliefs about CI soft surgery have changed since 2010 and shows areas of current consensus and disagreement.

## 1. Introduction

Experts now widely agree that steps should be taken to minimize the perioperative trauma caused during cochlear implant (CI) surgery [1,2]. Originally, this was done to try to preserve as much of the recipients’ low-frequency residual hearing as possible. The preservation was seen as increasingly important with the development of electric-acoustic stimulation (EAS) devices, which contain a hearing aid, which acoustically amplifies low frequencies, and a CI, which electrically stimulates middle and high frequencies. For an EAS device to be fully effective, users must have sufficient postoperative residual hearing to benefit from the acoustic stimulation and, thereby, achieve better hearing results than CI-only users [3,4,5,6]. It has, however, been increasingly recognized that “soft surgery techniques” should be used with every CI recipient, even if they are not an EAS candidate. Reasons for this include the following: avoiding trauma when possible, CI-only candidates having significantly better speech perception when their hearing is preserved, and preserving the inner ear structures, which could potentially allow the recipient to benefit from a technology that has not yet been developed [7,8].

Soft surgery in cochlear implantation was first advocated for by Lehnhardt [9]. Since then, his work has been built upon with much research and discussion on which techniques and technologies are most likely to achieve maximum atraumaticity (e.g., [10,11,12,13,14]). To gather expert opinion on this topic, O’Connor and O’Connor surveyed a large number of expert surgeons; their results were published in 2010 [14]. The study presented a clear picture of the state-of-the-art technique at that time; however, these findings may not be clinically relevant today. Thus, the primary aim of this study was to repeat (with a few modifications) the survey used by O’Connor and O’Conner [14] to determine what the expert opinion is now (i.e., 2020). This study will reveal how opinions have changed in the past decade and what remains to be clarified about this critical aspect of CI provision.

## 2. Methods

### 2.1. Survey Development

The survey was adapted from the original survey used in 2010, which consisted of 23 statements [14]. To update the content of the survey, 6 statements were deleted, and 8 new statements were added. The 6 deleted statements were as follows:The patient should have a pure-tone threshold in the quiet at 500 Hz of at least: 40 dB; 50 dB; 60 dB; 70 dB.The patient should have a monosyllable word score less than: 30%; 40%; 50%; 60%.The depth of insertion should be no deeper than: 10 mm; 15 mm; 18 mm; 21 mm.Should the electrode lie in the scala tympani laterally, in the mid scala, or next to the modiolus: laterally; mid scala; next to the modiolus?Following insertion of the electrode, viewed radiologically, it should not have passed around the cochlea more than: 360°; 320°; 280°; 240°.Would you say that you answered the questionnaire from clinical experience; knowledge from the literature/meetings; intuition?

These statements were removed from the current survey because they are of less relevance in current practice or because they were replaced with similar statements in the updated questionnaire. For example, pre-operative quantifications of residual hearing levels are not relevant because it is not believed that hearing-preserving techniques should always be applied. The statements regarding insertion depth and angle and where the electrode array should lie were replaced with new statements (#18–20).

The 8 statements that were added can be found in Table 2 (items 18–25, see Results section). The new questions assessed recent insights on EAS and hearing preservation since the original survey.

### 2.2. Dissemination, Completion, and Return of the Survey

Surgeons were sent an introductory email with the questionnaire to be completed. The survey was emailed to members of the HEARRING group, which is a worldwide collaborative of more than 30 comprehensive hearing centers dealing with all aspects of hearing restoration with implantable devices. Members are committed to leading research in hearing implant science, advancing audiological procedures, and developing and improving surgical techniques [15].

Questionnaire responses were not anonymized; however, responses were analyzed as an aggregate rather than on an individual level. Returning the completed questionnaire was taken as implied consent.

### 2.3. Statistics

For each statement, the absolute and the relative frequency are presented. Missing data are treated as missing values. All returned questionnaires were included, i.e., no amount of missing answers invalidated the questionnaire.

## 3. Results

### 3.1. Response

Questionnaires were sent to surgeons at 31 clinics. Eighteen questionnaires were returned. Responding surgeons practiced in 10 different countries: Australia, Austria, Belgium, Canada, Germany, Japan, Spain, Switzerland, the UK, and the USA.

### 3.2. Results Per Statement

See Table 1 and Table 2 for the results of each questionnaire item. Table 1 shows a comparison of responses to the items contained in the original survey and the present survey; Table 2 shows the items that were contained only in the present survey.

## 4. Discussion

Modern CI surgery prioritizes “soft surgery”, i.e., the surgical techniques and materials that are most likely to minimize inner ear trauma. Pioneered by Lehnhardt [9], soft surgery has been the subject of much research and continual refinement. In 2010, O’Connor and O’Connor surveyed over 60 CI surgeons regarding their beliefs about aspects of soft surgery [14]. The results of this study provided valuable information about what experts believed was state of the art at that time. We slightly modified that survey and used it to gain an updated perspective of soft surgical materials and techniques from a group of 18 experienced CI surgeons from around the world. In this section, we highlight what their answers reveal about state of the art now (in 2020), where broad agreement has been reached, and where consensus is lacking.

Present respondents placed more importance on preoperative radiological measurements of the cochlea (item 1) compared to past respondents. While the percentage of “very important” was almost unchanged (34.4% in 2010 vs. 33.3% in 2020), the percentage of “somewhat important” increased by 24.1%, and the percentage for “not important at all” decreased by 12.1%. This reflects the increased recognition that a one-size-fits-all approach is probably suboptimal and that cochlear size is very likely an important factor in the preservation of residual hearing and speech outcomes [17,18,19].

How deep the electrode array should be inserted (item 22) was not addressed in the original survey, so it is not possible to show how opinions have changed over time. However, results in the present survey are clear: 87.7% of present respondents answered that insertion depth is “somewhat important” or “very important”. This may reflect that insertion depth is related to array choice and the amount of residual hearing each recipient has. In short, the more residual hearing the recipient has, the shorter the array they should receive [1], although other factors may also influence this decision (e.g., progressive hearing loss, dysfunctional residual hearing, or an inability to wear an EAS ear mold).

Achieving the proper depth of insertion is related to the size of each individual’s cochlea. This is so that the electrode contacts on the array can be tonotopically matched to their location on the individual’s cochlea. Respondents increasingly believed that the frequency allocation of the electrode arrays should reflect their Greenwood location: 72.3% of present respondents rated item 17 as important, whereas, in 2010, only 37.7% of respondents felt that this was important or very important. No respondents rated the item as unimportant.

Some experts advocate partial electrode insertion (e.g., [20]) while others argue for complete insertion [21,22]. Deep insertion has the advantage of allowing better electrical hearing if the recipient loses their residual hearing (however long after surgery), although shallow insertions may lead to better hearing preservation rates [18,23].

An additional argument in favor of deep insertion with a long array is that because some hearing loss is progressive over time, EAS users may, with time, become CI-only users. Thus, deep insertion in such cases removes the need for a revision implantation. Furthermore, if residual hearing could not be preserved postoperatively, a deep insertion will be of benefit to the recipient (who would then be a CI-only user). To this end, since surgeons cannot guarantee that residual hearing can be preserved, EAS candidates (similarly to all CI candidates) should be carefully counseled regarding possible postoperative outcomes. Regardless of their view, advocates of deep and shallow insertion agree that insertion depth is important.

Regarding electrode arrays and how and where to insert them, the respondents in the present survey were clearly in favor of slow insertion (item 15) via the round window (RW) (item 5) when possible. Respondents in 2010 advocated RW insertion (55.7%) and slow insertion (78.7%); however, now all or almost all of respondents (94.4% and 100% on items 5 and 15, respectively) regard them as important (combined scores of “somewhat important” and “very important”). The general preference for the RW over cochleostomy is likely due to research finding that insertion via the RW is less likely to cause trauma and more likely to lead to better speech outcomes [10,21,24,25], although these results are far from unanimous (e.g., [26,27,28]). To access the RW, respondents indicated that the membrane should be incised with a sharp instrument (now 100% important, previously 83.6% important on item 10). Furthermore, the likelihood of trauma is minimized when the insertion is done slowly [29,30]. This has increasingly been recognized (item 15); whereas in the previous survey only 55.7% of respondents answered “very important”, in the present survey, 94.4% answered “very important”.

Respondents indicated a preference for flexible lateral wall arrays (items 18 and 21). This is consistent with recent findings that flexible lateral wall arrays enable atraumatic insertion and, therefore, hearing preservation [3,31,32]. However, this belief is not uncontested (e.g., [33,34]).

Items 4, 6, 7, 11, 12, 13, and 16 are all cochleostomy specific. As this is no longer the participants’ insertion site of choice, we simply highlight what changes this survey has revealed. Regarding steroid or hyaluronic acid (Healon) use, giving a single intravenous dose of corticosteroid prior to cochleostomy (item 4) is now regarded as very important (88.2% currently; 27.9% previously). Placing a drop of steroid solution at the cochleostomy site (item 11) is now regarded as more important (77.8 currently; 36.1% previously), while placing a drop of hyaluronic acid (Healon) to cover the cochleostomy is regarded as less important (22.3% currently; 62.7% previously). Lastly, that the cochleostomy should be smaller than 1 mm is now regarded as less important (56.6% currently; 85.2% previously), which could be explained as a manifestation of concerns regarding a potential rise in intracochlear pressure with smaller openings.

Most respondents (76.4%) indicated that dedicated EAS surgery training is important (item 23; 0% believed it unimportant), a finding that validates the continued existence of teaching centers. Over 70% of participants also believed that dedicated audiological rehabilitation training (item 24) is important.

Lastly, we found it interesting that 12/18 (66.7%) respondents answered that using the formula for calculating residual hearing postoperatively set forth by Skarzynski et al. [16] is important or very important. While 66.7% is certainly a majority, 16.7% were neutral and 16.7% felt its use is unimportant or very unimportant. Whether these latter 33.3% of respondents perceive a flaw in the formula or whether they simply do not see the need for such a formula would be interesting to discover.

A summary of what is regarded as more or less important now compared to in 2010 and of where general agreement does and does not exist can be found in Table 3. For convenience and clarity, “somewhat important” and “very important” scores were here combined as “important”; similarly, “somewhat unimportant” and “not important at all” scores were combined as “not important”.

An item was considered to be more important now or less important now if the percentage of “important” or “not important” responses changed by ±20 percentage points between the previous survey and this survey (e.g., the response to item 1). An item was considered to be in general agreement when there was a response of ≥80% in any one answer category (“important”, “not important”, or “neutral”). Lack of general agreement was considered to exist when no single answer category achieved < 66.6%. (Table 3 here).

This study has some limitations. Firstly, this survey did not address other potentially important factors in preserving residual hearing, e.g., age of recipient; etiology of hearing loss, including genetic testing; or the presence of any anatomical irregularities. Secondly, the present questionnaire was answered by 18 respondents, compared to the 61 who answered the original questionnaire. Further, none of the 18 respondents who responded in the 2020 questionnaire also participated in original questionnaire. Thus, it is not possible to track individual changes in belief over time. Lastly, all 18 of these respondents are part of a single working group (HEARRING) and share the aim of improving every aspect of cochlear implantation.

In conclusion, expert opinion is influenced by personal experience, peer-group discussion, and the current best available scientific knowledge. As such, it is anticipated that opinions will evolve over time, with the ultimate goal of reaching a strongly evidenced consensus on best practice.

The results of this survey make it clear that the respondents favor round window insertion with an atraumatic electrode array. Experts believe that insertion should be performed slowly, insertion depth is important, and systematic antibiotics should be given peri- and postoperatively. To this end, dedicated soft surgery training is important. Although steroid use is considered to be very important, the mode and timing of delivery remain contentious. This survey provides a contemporary summary of the “state of the art” of CI surgery with attempted hearing and structural preservation, highlighting technical factors of fundamental importance and areas in need of further research effort.

## Figures and Tables

**Table 1 jpm-12-01551-t001:** Survey responses for each item that appears in both surveys. “Old” refers to the original survey (see ref. 15); “new” refers to the present survey. Note: the items were not numbered in either survey; numbers have been added in this paper for convenience of reporting.

		Survey	Not Important at All	Somewhat Unimportant	Neutral	Somewhat Important	Very Important	Not Sure
1	As cochleae are of different sizes, radiological measurement of the size is needed.	Old	14 (23.2%)	5 (8.2%)	10 (16.4%)	9 (14.8%)	21 (34.4%)	2 (3.3%)
		New	2 (11.1%)	1 (5.6%)	2 (11.1%)	7 (38.9%)	6 (33.3%)	0 (0%)
2	Systematic antibiotics should be given peri- and postoperatively.	Old	4 (6.6%)	3 (4.9%)	4 (6.6%)	16 (26.2%)	32 (52.5%)	2 (3.3%)
		New	0 (0%)	1 (5.9%)	0 (0%)	4 (23.5%)	12 (70.1%)	0 (0%)
3	The middle ear cavity should be flushed with a non-aminoglycoside antibiotic solution.	Old	23 (37.7%)	7 (11.5%)	8 (13.1%)	15 (24.6%)	7 (11.5%)	1 (1.6%)
		New	7 (38.9%)	1 (5.6%)	2 (11.1%)	2 (11.1%)	5 (27.8%)	1 (5.6%)
4	Prior to cochleostomy, give a single intravenous dose of corticosteroid.	Old	4 (6.6%)	9 (14.8%)	9 (14.8%)	20 (32.8%)	17 (27.9%)	2 (3.3%)
		New	0 (0%)	0 (0%)	0 (0%)	2 (11.8%)	15 (88.2%)	0 (0%)
5	Introduce the electrode array through the round window if possible.	Old	7 (11.5%)	9 (14.8%)	8 (13.1%)	21 (34.4%)	13 (21.3%)	3 (4.9%)
		New	0 (0%)	0 (0%)	1 (5.6%)	2 (11.1%)	15 (83.3%)	0 (0%)
6	OR/IF NOT: The cochleostomy should be centered approximately 1 mm anterior and inferior to the horizontal midline of the round window.	Old	1 (1.6%)	0 (0%)	6 (9.8%)	15 (24.6%)	38 (62.3%)	1 (1.6%)
		New	1 (6.3%)	0 (0%)	2 (12.5%)	4 (25.0%)	8 (50.0%)	1 (6.3%)
7	The cochleostomy should be small, no more than 1 mm in diameter.	Old	1 (1.6%)	3 (4.9%)	5 (8.2%)	23 (37.7%)	29 (47.5%)	0 (0%)
		New	1 (6.3%)	0 (0%)	6 (33.3%)	1 (6.3%)	8 (50.0%)	0 (0%)
8	Expose the endosteum, but keep it intact.	Old	0 (0%)	5 (8.2%)	1 (1.6%)	21 (34.4%)	34 (55.7%)	0 (%)
		New	0 (0%)	0 (0%)	3 (16.7%)	3 (16.7%)	10 (58.8%)	1 (5.6%)
9	Flush bone dust from the mastoid and middle ear with Ringer’s solution.	Old	2 (3.3%)	3 (4.9%)	5 (8.2%)	15 (24.6%)	36 (59.0%)	0 (0%)
		New	1 (5.6%)	0 (0%)	2 (11.1%)	8 (44.4%)	7 (38.9%)	0 (0%)
10	Incise the endosteum or round window membrane using a sharp instrument.	Old	1 (1.6%)	2 (3.3%)	7 (11.5%)	20 (32.8%)	31 (50.8%)	0 (0%)
		New	0 (0%)	0 (0%)	0 (0%)	5 (27.8%)	13 (72.2%)	0 (0%)
11	Place a drop of steroid solution at the cochleostomy site.	Old	10 (16.4%)	12 (19.7%)	13 (21.3%)	15 (24.6%)	7 (11.5%)	4 (6.6%)
		New	0 (0%)	1 (5.6%)	3 (16.7%)	5 (27.8%)	9 (50.0%)	0 (0%)
12	Place a drop of hyaluronic acid (Healon) to cover the cochleostomy.	Old	6 (9.8%)	4 (6.6%)	13 (21.3%)	20 (32.8%)	17 (27.9%)	1 (1.6%)
		New	8 (50.0%)	2 (11.1%)	2 (11.1%)	3 (16.7%)	1 (5.6%)	0 (0%)
13	There should be minimum suction at the cochleostomy site.	Old	0 (0%)	1 (1.6%)	0 (0%)	6 (9.8%)	54 (88.5%)	0 (0%)
		New	0 (0%)	0 (0%)	0 (0%)	1 (5.6%)	17 (94.4%)	0 (0%)
14	Use a sheath or insertion tube to avoid contamination of the electrode array by blood or bone dust.	Old	5 (8.2%)	12 (19.7%)	18 (29.5%)	15 (24.6%)	8 (13.1%)	3 (4.9%)
		New	6 (33.3%)	1 (5.6%)	3 (16.7%)	4 (22.2%)	3 (16.7%)	1 (5.6%)
15	The electrode array should be inserted slowly.	Old	0 (0%)	1 (1.6%)	10 (16.4%)	14 (23.0%)	34 (55.7%)	2 (3.3%)
		New	0 (0%)	0 (0%)	0 (0%)	1 (5.6%)	17 (94.4%)	0 (0%)
16	Following insertion, immediately seal the cochleostomy with connective tissue and fibrin glue.	Old	2 (3.3%)	3 (4.9%)	6 (9.8%)	15 (24.6%)	35 (57.4%)	0 (0%)
		New	0 (0%)	2 (11.1%)	3 (16.7%)	8 (44.4%)	4 (22.2%)	1 (5.6%)
17	The frequency allocation to the electrodes should reflect their Greenwood location in the cochlea.	Old	4 (6.6%)	5 (8.2%)	24 (39.3%)	11 (18.0%)	12 (19.7%)	5 (8.2%)
		New	0 (0%)	0 (0%)	2 (11.1%)	10 (55.6%)	3 (16.7%)	3 (16.7%)

Using a 5-point Likert scale with the additional option of answering “not sure”, participants were asked to rank the importance of various aspects of EAS as “not important”, “somewhat important”, “neutral”, “somewhat important”, “very important”, and “not sure”. Responses did not carry a numerical value.

**Table 2 jpm-12-01551-t002:** Survey responses for the 8 new items. Note: the paper referenced in item #25 is reference [16] in this paper.

		n	Not Important at All	Somewhat Unimportant	Neutral	Somewhat Important	Very Important	Not Sure
18	Use of lateral wall electrodes	18	0 (0%)	0 (0%)	2 (11.1%)	3 (16.7%)	12 (66.7%)	1 (5.6%)
19	Use of midscalar electrode	16	5 (31.3%)	1 (6.3%)	5 (31.3%)	0 (0%)	1 (6.3%)	4 (25.0%)
20	Use of peri-modiolar electrode	15	4 (26.7%)	5 (33.3%)	4 (26.7%)	0 (0%)	0 (0%)	2 (13.3%)
21	Electrodes being very flexible	17	0 (0%)	0 (0%)	0 (0%)	3 (17.6%)	14 (82.4%)	0 (0%)
22	Insertion depth of the electrode	17	0 (0%)	1 (5.9%)	1 (5.9%)	3 (17.6%)	12 (70.1%)	0 (0%)
23	Dedicated EAS surgery training	17	0 (0%)	0 (0%)	3 (17.6%)	4 (23.5%)	9 (52.9%)	1 (5.9%)
24	Dedicated audiological rehab training	17	0 (0%)	0 (0%)	2 (11.1%)	2 (11.1%)	11 (61.1%)	3 (16.7%)
25	The use of the ‘S’ calculation of the residual hearing ref Skarzynski et al. Acta Otolaryng 2013 [16]	18	2 (11.1%)	1 (5.6%)	3 (16.7%)	9 (50.0%)	3 (16.7%)	0 (0%)

**Table 3 jpm-12-01551-t003:** A list of items that are regarded as more important or less important now (2020) compared to the 2010 survey and items for which there exists general agreement or a lack of general agreement. The Skarzynski et al. paper at the end of “Lack of general agreement” is reference [16] in this paper.

*More important now* As cochleae are of different sizes, radiological measurement of the size is needed (item 1). Prior to cochleostomy, give a single intravenous dose of corticosteroid (item 4). Introduce the electrode array through the round window if possible (item 5). Place a drop of steroid solution at the cochleostomy site (item 11). The electrode array should be inserted slowly (item 15). The frequency allocation to the electrodes should reflect their Greenwood location in the cochlea (item 17). *Less important now* The cochleostomy should be small, no more than 1 mm in diameter (item 7). Place a drop of hyaluronic acid (Healon) to cover the cochleostomy (item 12). *Note: items 18–25 are new to this survey and, therefore, cannot be designated as more or less important.* *General agreement* Systematic antibiotics should be given peri- and postoperatively (item 2). Prior to cochleostomy, give a single intravenous dose of corticosteroid (item 4). Introduce the electrode array through the round window if possible (item 5). The cochleostomy should be centered approximately 1 mm anterior and inferior to the horizontal midline of the round window (item 6). Expose the endosteum, but keep it intact (item 8). Flush bone dust from the mastoid and middle ear with Ringer’s solution (item 9). Incise the endosteum or round window membrane using a sharp instrument (item 10). Place a drop of steroid solution at the cochleostomy site (item 11). There should be minimum suction at the cochleostomy site (item 13). The electrode array should be inserted slowly (item 15). Use of lateral wall electrodes (item 18). Electrodes being very flexible (item 21). Insertion depth of the electrode (item 22). Dedicated EAS surgery training (item 23). *Lack of general agreement* The middle ear cavity should be flushed with a non-aminoglycoside antibiotic solution (item 3). The cochleostomy should be small, no more than 1 mm in diameter (item 7). Place a drop of hyaluronic acid (Healon) to cover the cochleostomy (item 12). Use a sheath or insertion tube to avoid contamination of the electrode array by blood or bone dust (item 14). Use of midscalar electrode (item 19). The use of the ‘S’ calculation of the residual hearing ref Skarzynski et al. 2013 (item 25) [16].

## Data Availability

Not applicable.

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
