# Peer review of "Minimally Traumatic Cochlear Implant Surgery: Expert Opinion in 2010 and 2020"

_jpm, 2022, doi:10.3390/jpm12101551_

Round 1

Author Response

Thank you for the time you invested in reviewing the manuscript & for the compliments.

Reviewer 2 Report

This article is very relevant as cochlear implant science and surgery keeps evolving. Many newer aspects have now come into practice in last ten years though the basic tenet of surgery remains the same. Authors efforts to evaluate best surgical practices among surgeons in recent time is a good one and deserves wider dissemination.

Author Response

Thank you for the time you invested in reviewing the manuscript & for the compliments

Reviewer 3 Report

Overall, the authors present a nice review and study gauging expert opinion on a pertinent topic — maximizing hearing preservation in cochlear implant surgery. Although no real statistical analysis was provided, authors qualitatively explained their findings very well. However, there are some suggestions that this reviewer has that may strengthen the manuscript. 

1. Were some of the participants in the study the same? If so, could the authors performed paired comparisons between their answers in 2010 compared to present day?

2. The fact that many of the participants may not be the same is a significant limitation—this should be made clear in the limitations section. 

Author Response

Thank you for the review. Regarding your comments:

  1. None of the participants in our study contributed to the original study in 2010
  2.  We have added this as a limitation in the Discussion